# Defense Strategies of Rice in Response to the Attack of the Herbivorous Insect, *Chilo suppressalis*

**DOI:** 10.3390/ijms241814361

**Published:** 2023-09-21

**Authors:** Xing Xiang, Shuhua Liu, Hongjian Li, Andrews Danso Ofori, Xiaoqun Yi, Aiping Zheng

**Affiliations:** State Key Laboratory of Crop Gene Exploration and Utilization in Southwest China, Sichuan Agricultural University, Chengdu 611130, China; weileshimo@163.com (X.X.); shuhualiu618@163.com (S.L.); 15708179908@163.com (H.L.); 2019511003@stu.sicau.edu.cn (A.D.O.); yxquan1981@163.com (X.Y.)

**Keywords:** *Chilo suppressalis*, rice, defense strategies, transgenic approaches, rice insect resistance

## Abstract

*Chilo suppressalis* is a notorious pest that attacks rice, feeding throughout the entire growth period of rice and posing a serious threat to rice production worldwide. Due to the boring behavior and overlapping generations of *C. suppressalis*, the pest is difficult to control. Moreover, no rice variety with high resistance to the striped stem borer (SSB) has been found in the available rice germplasm, which also poses a challenge to controlling the SSB. At present, chemical control is widely used in agricultural production to manage the problem, but its effect is limited and it also pollutes the environment. Therefore, developing genetic resistance is the only way to avoid the use of chemical insecticides. This article primarily focuses on the research status of the induced defense of rice against the SSB from the perspective of immunity, in which plant hormones (such as jasmonic acid and ethylene) and mitogen-activated protein kinases (MAPKs) play an important role in the immune response of rice to the SSB. The article also reviews progress in using transgenic technology to study the relationship between rice and the SSB as well as exploring the resistance genes. Lastly, the article discusses prospects for future research on rice’s resistance to the SSB.

## 1. Introduction

“Bread is the staff of life”, and so is rice (*Oryza sativa* L.), as the staple food for half of the world’s population [1]. This important food crop greatly affects people’s basic needs, playing an important role in the country’s economic security and social stability. With the increase in the world’s population, the consumption trend of rice is also constantly increasing, and it is expected that the demand for rice will be higher in the future. Based on economic analysis, when the per capita consumption level in Asia and Africa continues to rise, the growth of rice consumption may even exceed the growth of population [2]. However, research reports have shown that crop diseases and pests cause global crop yield losses of 20~40%, with most of these losses caused either directly or through diseases transmitted by pests [3]. Therefore, cultivating high-yield and insect-resistant rice varieties to ensure the sustainable development of the rice industry is currently an urgent problem to be solved.

*Chilo suppressalis* (Lepidoptera: Pyralidae), also known as rice stem borer, inflicts damage throughout the entire growth and development period of rice, as well as on crops such as *Zizania latifolia*, wheat, and corn [4]. *C. suppressalis* is one of the three main pests of the Lepidoptera in the rice ecosystem [5], causing significant harm to agricultural production and resulting in substantial losses in rice yield [6]. This translates to an annual economic loss of approximately CNY 11.5 billion [7]. In recent years, the harm caused by this pest has been increasing due to global warming, which has led to higher temperatures in winter and spring, along with changes in farming systems.

*C. suppressalis* is a holometabolous insect that undergoes four stages throughout its life cycle, namely, the egg stage, the larval stage, the pupal stage, and the adult stage. The egg is a flat oval shape, consisting of 10 to over 100 egg masses arranged in a fish scale shape. The eggs are mainly distributed at the base of the leaf back, near the leaf sheath, and some are also laid near the leaf tip, toward the front of the leaf. The egg is milky white when laid and gray-black at hatching. The larvae have five brown longitudinal lines on their back and a gray-white belly. The width of the head shell is used as a morphological indicator for larval division, which allows them to be categorized into seven different larval stages [8]. The first hatched larvae cluster within the leaf sheath, leading to withered sheaths. After the third larval stage, which occurs approximately 10–12 days after the larvae hatch, the larvae bore into the stem and cause damage. This damage results in withered seedlings during the tillering stage of rice, withered booting ears during the booting stage, white ears during the heading stage, and insect damage during the mature stage. When symptoms such as white spikes appear in the crop, it can be determined by observation that the first leaf from the base of the rice plant is the “healthy green leaf” to be harmed by the striped stem borer. However, the striped stem borer mainly harms the middle of the rice field, while the pink stem borer mainly affects the surrounding areas. When the temperature rises above 11 °C, the mature larvae undergo pupation within the stem or between the leaf sheaths and the stem. The pupa is yellowish-brown, with five brown vertical lines visible on the back in the early stage, with the middle three being more prominent. When the temperature exceeds 15–16 °C, adults emerge normally. After emerging, adults remain hidden in the lower part of the rice plant during the day and become active in flight during the night. They predominantly engage in mating activities before midnight. Following successful mating, they begin laying eggs at intervals of 1–2 days. The highest abundance of egg laying occurs between 20:00 and 21:00. A female moth can lay an average of 5–6 egg masses, laying 200–700 eggs during its lifetime. Generally, there can be approximately two generations per year, with mature larvae or pupae overwintering on hosts such as rice stubble or straw. Figure 1 provides a schematic description of a complete generation cycle.

Currently, due to the lack of germplasm resources with high resistance to the rice stem borer, the main method of controlling the SSB still relies heavily on the use of chemical insecticides [9]. However, the excessive use of chemical insecticides has led to insect resistance, diminishing the effectiveness of these insecticides. Additionally, it has led to a decline in the population of natural enemies that prey on crop pests. The long-term use of chemical insecticides has also raised concerns about environmental pollution and food safety issues, which are becoming more prevalent [10]. With the increasing difficulty and cost of developing new pesticides, it has also had a negative impact on economic development. How to effectively prevent and control pests while ensuring food and environmental security is the main challenge that is currently being faced. Therefore, it is particularly important to gain a deeper understanding of the interaction mechanism between rice and *C. suppressalis* in order to quickly find effective pest control methods that can replace the use of chemical insecticides, thereby achieving sustainable crop production.

To date, a large amount of cutting-edge research on the resistance of rice to herbivorous insects mainly focuses on the interaction between rice and the brown planthopper [11,12,13,14,15,16]. The interaction between rice and *C. suppressalis* has not been thoroughly and comprehensively studied. This article will review research progress on the occurrence of the SSB, the genetic resistance of rice, and transgenic methods to achieve rice resistance to the SSB.

## 2. Defense Strategies of Rice against the Herbivorous Insect, *C. suppressalis*

The earliest terrestrial plants on Earth appeared about 400 million years ago, while the earliest insects originated about 300 million years ago. This long-term co-evolutionary relationship has made plants the main food source for insects for a significant period. During this extended process of co-evolution, plants have developed a series of defense strategies against herbivorous insects. We will discuss these defense mechanisms from three perspectives: constitutive defense response, tolerant defense response, and induced defense response (Figure 2).

### 2.1. Constitutive Defense Response

Constitutive resistance refers to insect resistance characteristics that exist in plants before they are harmed by herbivorous insects. These characteristics play an important role in protecting plants from insect feeding [17,18]. Constituent defense is present throughout the entire life cycle of plants. Plants employ their own morphological and biochemical characteristics to defend themselves against harm caused by herbivorous insects. These characteristics include sharp and thick trichomes, thick epidermal wax, and secondary metabolites [19,20,21,22]. It is noteworthy that some constitutive defense mechanisms are greatly enhanced when plants are invaded [23,24,25,26]. Currently, rice lacks germplasm resources with high resistance to the SSB. Nevertheless, rice has some agronomic traits that provide some degree of protection against the damage caused by the SSB. For instance, some varieties have tightly wrapped leaf sheaths, while others have thick parietal cells in their stems. These traits discourage the invasion of newly hatched SSB larvae. Research indicates that under low nitrogen (0.3 mM nitrate) supply, the accumulation of lignin in rice significantly increases, thereby enhancing the constitutive defense against SSB infection [27]. Further research is needed to explore this form of defense against the SSB.

### 2.2. Tolerant Defense Response

Pest tolerance refers to plants’ ability to compensate for yield losses caused by pests. They achieve this by improving photosynthesis and enhancing nutrient absorption capacity after being eaten by herbivorous insects [28,29,30]. Rice plants, especially high-tillering varieties, possess the capability to compensate for pest damage. For instance, research has shown that over 75% of rice seedlings can be damaged by black rice stem flies without suffering yield losses [31]. After being damaged by the SSB, resistant rice varieties can compensate for the loss of stems by producing more tillers. Additionally, since withered tillers are shorter than the surrounding healthy tillers, the healthy tillers enjoy better ventilation and lighting conditions, thereby increasing the photosynthetic capacity of the surrounding healthy tillers. These two factors partly mitigate the impact of SSB damage on the normal growth of rice [32]. However, this compensatory ability weakens as plants mature, evidenced by tolerance indicators such as plant weight loss, leaf yellowing, and reduced yield [33,34]. Currently, the genetic basis of tolerance has not been thoroughly elucidated.

### 2.3. Induced Defense Response

The interaction between plants and herbivorous insects can be divided into three processes: contact, feeding, and oviposition. Each process of herbivorous insects triggers plant defense responses [35]. The generation of induced resistance is a complex physiological and biochemical process. It usually starts with the recognition of herbivore-associated molecular patterns (HAMPs) by plants, followed by the activation of a mitogen-activated protein kinase (MAPK) signal cascade as well as the activation of plant hormone signaling pathways such as jasmonic acid (JA), salicylic acid (SA), and ethylene (ET). Ultimately, this leads to the regulation of defense gene expression and defense compound synthesis [36]. The induced defense response exhibits characteristics of being more flexible and rational in allocating plant energy resources.

#### 2.3.1. Herbivore-Associated Molecular Patterns

It is widely accepted that plant-induced defense responses begin with the recognition of HAMPs or damage-associated molecular patterns (DAMPs) by plant pattern recognition receptors (PRRs). In recent years, various HAMPs have been identified in herbivorous insects, mainly from their oral secretions [35,37,38,39,40,41], microorganisms carried by insects [42], secretions from egg-laying insects [39,43,44,45,46], insect excrement [47,48,49], and volatiles released by insects [50,51]. Currently, only leucine-rich repeat receptor-like kinases (LRR-RLKs) in rice have been proven to play a significant role in rice recognition and defense against the SSB [52]. Unfortunately, there have been no reports thus far on the HAMPs involved in rice defense responses induced by the SSB.

#### 2.3.2. Activation of Defense Signal Transduction in Rice

It is widely believed that when plants perceive the damage caused by herbivorous insects, they induce changes in transmembrane ion flow, causing changes in the voltage striding over the membrane (Vm) on the cell membrane surface, cascade activation of mitogen-activated protein kinase (MAPK), bursts of reactive oxygen species (ROS), and further regulating the synthesis of plant hormones such as JA, SA, and ET [53].

After plants recognize HAMPs or DAMPs, they trigger depolarization and Vm changes in the plasma membrane within a few seconds, which causes the influx of Ca^2+^ into the apoplasmic and changes in transmembrane ion flows such as H^+^ and Cl^-^ within a few minutes [54,55]. The change in Ca^2+^ concentration in the plant cytoplasm is considered a calcium signal in early plant defense events. The information carried by Ca^2+^ needs to be decoded and transduced by Ca^2+^ binding proteins, known as Ca^2+^ sensors, in order to activate downstream defense reactions [56,57,58]. Changes in rice membrane potential and Ca^2+^ current caused by SSB feeding has not been researched yet.

The outbreak of ROS was first discovered in potatoes infected with *Fusarium oxysporum* in 1983 [59]. As a signaling molecule, it plays an important role in plant responses to abiotic and biotic stress. ROS is found in chloroplasts, mitochondria, peroxisome, and their plasma membranes [60]. In rice, the deletion of the mitochondrial outer membrane protein 64 (OM64) gene, located on the outer membrane of the mitochondria, constitutively activates the H_2_O_2_ pathway. This ultimately confers resistance on BPH, independently of SSB resistance [61]. H_2_O_2_ is not believed to be related to anti-chewing herbivores [62].

The MAPKs cascade is highly conserved and exists in almost all eukaryotes [63]. Both herbivorous insect damage and mechanical damage can quickly activate the MAPK signaling cascade, thereby transmitting plant defense responses [17,64]. The defense of rice against the SSB requires the participation of MAPKs. Silencing the *OsMPK3* gene reduces the JA level triggered by the SSB. This decreases the level of trypsin protease inhibitors (TrypPIs) induced by herbivores, which improves the performance of the SSB larvae [65]. Under SSB attack, the ethylene-responsive factor gene, *OsERF3*, positively regulates the transcription levels of *OsMPK3* and *OsMEK3* as well as the expression levels of downstream transcription factor genes, *OsWRKY53* and *OsWRKY70*, in rice. However, it has no positive regulatory effect on the transcription level of *OsMPK6* [66]. *OsLRR-RLK1* acts upstream of the MAPK cascade, positively regulating the expression of *OsMEK4*, *OsMPK3*, and *OsMPK6*, as well as downstream defense-related WRKY transcription factors, thereby endowing rice with resistance to the SSB [52]. In addition, *OsWRKY53* acts as a negative feedback regulator for MPK3/MPK6, thereby serving as an early inhibitor of SSB-induced defense in rice [67].

After sensing the harm caused by herbivorous insects and triggering early defense events such as the MAPKs cascade, plants proceed to regulate the synthesis of related signaling molecules and activate defense signal transduction pathways. Among these pathways, the jasmonic acid signaling pathway assumes a central role in the defense response induced by plant pests. In the JA signaling pathway, jasmonic acid and its precursors such as OPDA as well as derivatives such as jasmonic acid isoleucine (JA-Ile), are key substances involved in the defense response against insects [68]. The JA signaling pathway is specifically illustrated as follows: when plants suffer damage from phytophagous insects, α-linolenic acid (18:3) is released by galactolipids in the plasma membrane under the action of acyl lipohydrolase. The released α-linolenic acid undergoes oxidation through 13-lipoxygenase, leading to epoxidation and cyclization reactions, which generate OPDA. OPDA is then transported to the peroxisome, where it undergoes one round of reduction and three rounds of β-oxidation to form (+)-7-iso-JA. In the cytoplasm, (+)-7-iso-JA conjugates with isoleucine under the influence of JAR1, a jasmonate amino acid conjugate synthase [69,70,71]. JA-Ile is a typical functional molecule with JA biological activity in plants. It is transported to the nucleus by the ATP-binding cassette (ABC) transporter JAT1, and has the ability to bind complex receptors composed of COI1 (coronatine insensitive 1), JAZ (jasmonate-ZIM-domain), and inositol polyphosphate cofactor. COI1 serves as a component of the SKP1-CUL1-F-box protein E3 ubiquitin ligase (SCF^COI1^) complex. Upon binding with JA-Ile, it facilitates the degradation of JA’s inhibitor JAZ through the 26S proteasome [72,73]. The ubiquitination and hydrolysis of JAZ can relieve the inhibition of *MYC2* and its associated transcription factors, *MYC3*, *MYC4*, and *MYC5*, enabling the expression of defense genes. This, in turn, leads to the development of broad-spectrum resistance in plants against herbivorous insects [74]. In recent years, significant research has been conducted on the key genes and chemicals involved in the JA synthesis pathway in relation to insect resistance responses in rice. Transcriptome analysis and analysis of defense-related chemicals in rice exposed to the SSB have revealed that jasmonic acid (JA), salicylic acid (SA), and ethylene are the primary hormones involved in the defense response triggered by the SSB in rice [75]. Research has revealed that during SSB attack, four genes involved in JA biosynthesis (*OsLOX9*, *OsJAR1*, *OsDAD1*, and *OsAOC*) and five genes involved in JA signaling (*OsLOXL-2*, *OsAOS3*, *OsJAZ1*, *OsJAZ7*, and *OsJAZ9*) in rice were significantly upregulated. Additionally, JA levels exhibit a substantial increase. However, under dual attack of BPH and SSB, JA levels are found to be inhibited [76]. Oxylipins play a vital role in the JA signaling pathway, enabling plants to defend themselves against herbivorous insects. In its biosynthesis, 9-LOX is actively involved. Silencing the *Osr9-LOX1* gene in rice can enhance JA expression, increase the synthesis of TrypPIs, and positively regulate rice resistance to the SSB [77]. The overexpression of rice *AOC* (production of OPDA) and *OPR3* (reduction of OPDA) genes in ZH11 can enhance rice resistance to the SSB, suggesting that this enhanced resistance is not related to OPDA [78]. By analyzing the levels of JA and SA in ir-*lrr* rice and WT plants after SSB attack and mechanical damage, it can be concluded that *OsLRR-RLK1* does not regulate the production of trauma-induced JA, JA-Ile, and SA in the absence of SSB attack [52]. Healthy rice plants were exposed to SSB-induced rice volatiles, and two JA signaling genes (*DOX2* and *LOX8*) were directly induced, resulting in a significant increase in JA accumulation [79]. The absence of *OM64* in rice stimulates SSB-induced JA biosynthesis and response, enhancing the resistance of rice to the SSB [61]. Through silencing the rice gene *OsWRKY53*, it has been discovered that this gene acts as a negative regulator of the levels of JA, JA-Ile, and ET induced by the herbivorous SSB. As a result, this mediates the activity of TrypPIs and confers resistance on the SSB [67]. The antisense expression (as-*lox*) of the 13-lipoxygenase gene, *OsHI-LOX*, located in the rice chloroplast reduced the level of JA and trypsin protease inhibitor (TrypPI) induced by the SSB, improved the larval status of the SSB and rice leaf folder (LF), and increased the damage caused by the SSB and LF larvae [62]. The antisense expression of the rice hydrogen peroxide lyase gene, *OsHPL3*, showed enhanced induction of JA and trypsin inhibitor, providing better resistance to the SSB [80]. The ethylene-responsive factor gene, *OsERF3*, regulates the resistance of rice to the SSB by inhibiting MAPK inhibitors and regulating the JA pathway [66]. Silencing the expression of the *OsMPK3* gene reduces the level of JA induced by the SSB, thereby improving the survival status of the SSB [65]. Additionally, studies have shown that genes such as allene oxide synthase (AOS) and allene oxide cyclase (AOC) respond to SSB attacks on rice through the JA pathway [81,82]. In recent years, extensive research has been conducted on the JA pathway, and the understanding of the JA signaling pathway has gradually deepened. However, there are still many unresolved issues, such as the connection between early defense signals and acyl-lipid hydrolases that initiate JA biosynthesis, which require further exploration by researchers [83].

The SA-dependent signaling pathway regulates the expression of various defense response genes. Among these genes, the receptor NPR1 (non-expressor of PR genes1) of SA plays a crucial role in SA-mediated defense responses, enabling plants to achieve a broad spectrum of systemic acquired resistance [84]. Phenylalanine ammonia lyase (PAL) and isochorismate synthase (ICS) are two key enzymes involved in the biosynthesis of SA in plants [85,86]. It is generally believed that chewing mouthpiece insects primarily induce plant JA signaling pathways, while piercing-sucking mouthpiece insects induce a combination of plant JA, SA, and ET signaling pathways [87,88,89]. Through transcriptome analysis of rice 48 h after SSB attack, nine SA response genes (*OsPR2*, *OsPR4*, *OsPR4B*, *OsPR4C*, *OsPR4D*, *OsPR6*, *OsPR10*, *OsPR10*, and *OsPR10B*) were found to be activated by SSB infection [76]. In rice, the cytochrome P450 gene *CYP71A1* encodes tryptamine 5-hydroxylase, which catalyzes the conversion of tryptamine to 5-hydroxytryptamine. The *CYP71A1* functional deficient mutant (*CYP71A1*-KO) can make rice resistant to the SSB by inhibiting the synthesis of 5-hydroxytryptamine. However, the expression levels of two salicylic acid biosynthesis genes, *OsICS1* and *OsPAL*, decrease after SSB attack [16]. *OsLRR-RLK1* positively regulates rice resistance to the SSB, but its functional-deficient mutant accumulates higher SSB-induced SA levels [52].

When attacked by herbivorous insects, plants quickly activate the biosynthesis of ethylene to resist the harm caused by herbivorous insects [90]. The biosynthesis of ethylene begins with methionine, which can be catalyzed by S-AdoMet synthase (SAM) to generate S-AdoMet. It is further converted into ACC (1-amino cyclopropane-1-carboxylic acid) through ACC synthase (ACS). Under the action of ACC oxidase (ACOs), ACC generates ethylene, which can then be further perceived by the receptor. Currently, five ethylene receptors have been identified, namely, ETR1 (ethylene acceptor 1), ETR2 (ethylene acceptor 2), ERS1 (ethylene response sensor 1), ERS2 (ethylene response sensor 2), and EIN4 (ethylene sensing 4) [91,92]. In rice, the *OsERF3* positively regulates the expression of MAPK genes, *OsMEK3* and *OsMPK3*, and WRKY transcription factor genes, *OsWRKY53* and *OsWRKY70*. It also plays a role in regulating the synthesis of signal molecules, ET and JA, and the production of trypsin protease inhibitors to regulate the resistance of rice to the SSB [66]. Additionally, mutant lines with the silenced ACC synthase gene, *OsACS2*, in rice can reduce ethylene release. This reduction leads to a decrease in SSB-induced volatile organic compound synthesis and TrypPIs activity, ultimately resulting in the reduced resistance of rice to the SSB. However, external application of ACC can compensate for TrypPI activity and enhance rice resistance to the SSB [93].

#### 2.3.3. Defense Compounds

JA, SA, and ET signaling molecules can ultimately regulate the production of plant defense compounds. These compounds typically exert anti-insect effects by inhibiting nutrient and ion transport in insects, suppressing insect physiological metabolism, interfering with signal transduction in their bodies, and disrupting hormone-related physiological functions [94]. JA, SA, and ET signaling molecules can manifest as direct or indirect defenses in plants. They can result in the production of toxic secondary metabolites such as phenols [95], terpenoids [96], steroids [97], or alkaloids [98]. These compounds have the ability to directly kill insects and exhibit a broad-spectrum toxic effect on herbivorous insects. Plants can also produce defensive proteins, such as plant lectins [99], oxidase [100], and protease inhibitors [101], which weaken the ability of insects to digest food. Additionally, plants can modify the nutritional composition of the fed area to prevent pests from obtaining sufficient nutrition [102,103].

The biosynthesis of trypsin protease inhibitors in rice is regulated by JA, SA and ET signaling pathways. By tightly binding with the proteolytic enzymes in insects, the activity of proteolytic enzymes is inhibited, leading to indigestion and deficiency of essential amino acids in SSB, which can have adverse effects on growth and development. The positive participating genes are *OsMPK3* [65], *OsLRR-RLK1* [52], *OsHI-LOX* [62], *OsERF3* [66], and the reverse participating genes are *OsWRKY53* [67], *OsHPL3* [80], and *Osr9-LOX1* [77]. Furthermore, through the transcriptome sequencing analysis of cultivated rice 1688 and 1654 (resistant to the SSB) and cultivated rice 1665 (susceptible to the SSB), it has been found that the plant lectin gene is closely related to the resistance of rice to the SSB [104].

After herbivorous insects invade plants, in addition to triggering their defense reactions, plants often rely on external forces for protection. Herbivore-induced plant volatiles (HIPVs) induced by insect feeding are like “radio signals”. When parasitic wasps or predatory predators receive signals, they act like “shells” to achieve precise strikes on pests. This is the indirect defense of plants. HIPVs can be broadly categorized into three categories: green leaf volatile substances, terpenoids, and other substances. Green leaf volatiles (GLVs) are volatile fatty acid derivatives, such as vinyl aldehyde, vinyl alcohol, and vinyl acetate, consisting of six carbon elements. When rice is attacked by the SSB, it can release a significant amount of (Z)-3-hexene-1-ol, which serves as an attractant for natural enemy parasitic wasps that parasitize SSB larvae. Similarly, other substances emitted by rice, such as linalool (terpenoids) or ethyl benzoate, methyl salicylate-4, 8-dimethyl-1, 3, 7-nonanetriene (DMNT), can also attract the arrival of parasitic wasps, thereby participating in the defense of rice against the SSB [79,80].

#### 2.3.4. Applying Omics Techniques to Understand Host Defense

Omics is a widely used and practical tool that allows for the comprehensive analysis of genes, proteins, and metabolites in host plants in response to the attack of pests. It encompasses disciplines such as genomics, transcriptome, proteomics, and metabolomics. In a study conducted by Sun et al. [81], the authors employed a combination of suppression subtractive hybridization (SSH) and dot blot hybridization techniques to sequence the entire SSH library. Through this approach, they identified 39 expressed sequence tags in rice that were upregulated in response to SSB larval feeding, indicating their involvement in the plants’ stress response to insect infestations. These upregulated ESTs included genes such as rice allene oxide cyclase (AOC), terpene synthase (TPS), and four protease inhibitor (PI) genes. Zhou et al. [75] conducted a study to investigate the changes in the transcriptome and compounds of rice when invaded by the SSB. Their research revealed that SSB infection causes significant changes in the expression level of 4545 rice genes, accounting for about 8% of the genome. Particularly, genes responsible for plant hormone biosynthesis and signal transduction showed notable alterations. Jasmonic acid (JA), salicylic acid (SA), and ethylene were identified as the main hormones involved in the defense response of rice against the SSB. Several secondary signal transduction components, such as those involved in Ca^2+^ signal transduction and G protein signal transduction, as well as receptors and non-receptor protein kinases and transcription factors, were found to be involved in the SSB-induced response of rice. Moreover, SSB infectation results in the accumulation of defense compounds, including trypsin protease inhibitor (TrypPIs) and volatile organic compounds. Liu et al. [105] studied changes in the gene expression and metabolic processes in rice plants fed continuously with SSB larvae at different times (0, 24, 48, 72, and 96 h) using next-generation RNA sequencing and metabolomics techniques. The results showed that a total of 4729 genes and 151 metabolites were regulated differently when rice plants were damaged by SSB larvae. Further analysis indicates that defense-related plant hormones, transcription factors, shikimate ester-mediated and terpenoid-related secondary metabolisms are activated, while growth-related counterparts are inhibited by the feeding of the SSB. The activated defense is driven by the catabolism of energy storage compounds (such as monosaccharides), which also leads to an increase in the level of metabolites involved in the defense response of rice plants. In a separate study, Liu et al. [76] conducted transcriptome sequencing analysis on wild-type rice and rice subjected to SSB infection for 48 h. The analysis revealed differential expression of 12512 genes, of which 6533 were upregulated and 5979 were downregulated in response to SSB feeding. Among the upregulated genes, four were involved in JA biosynthesis, namely, *OsLOX9*, *OsJAR1*, *OsDAD1*, and *OsAOC*. These genes were significantly upregulated after SSB infection. Additionally, five genes associated with JA signaling transduction (*OsLOXL-2*, *OsAOS3*, *OsJAZ1*, *OsJAZ7*, and *OsJAZ9*) and nine SA responsive genes (*OsPR2*, *OsPR4*, *OsPR4B*, *OsPR4C*, *OsPR4D*, *OsPR6*, *OsPR10*, *OsPR10A*, and *OsPR10B*) were also activated upon SSB invasion.

## 3. Transgenic Strategies

The Integrated Pest Management (IPM) strategy encompasses various methods, including agricultural control, physical control, biological control, and chemical control, to effectively manage harmful organisms below the level of economic damage. Furthermore, cultivating new resistant varieties through genetic engineering or traditional breeding will contribute significantly to the sustainable development of agriculture. However, due to the lack of germplasm resources with high resistance to the SSB, traditional breeding has not yet achieved results. In recent years, many potential genes related to resistance against the SSB have been isolated and identified from rice (Table 1). In addition, exogenous genes have also been extensively used in rice breeding for resistance to the SSB through transgenic methods (Table 1). In this context, we summarize the types of genes used, their expression strategies, and their efficacy in providing resistance to *C. suppressalis*.

### 3.1. Signal Transduction Genes

When plants detect damage from herbivorous insects, they promptly initiate signaling pathways, including MAPKs, JA, SA, and ET, and ultimately generate defense compounds to counterattack [17,118,119]. Therefore, the expression of the main switch gene that manipulates the upstream defense response will be one of the best choices for endowing rice with resistance to the SSB. Studies conducted on the rice variety, xiushui11, show that *OsMPK3* positively regulates rice’s defense against the SSB by regulating JA outbreaks and TrypPI levels [65]. Silencing the transcription factor gene, *OsWRKY53*, enhances rice’s resistance to the SSB by activating JA or ET signaling pathways [67]. Research on the rice variety, xiushui110, showed that the ethylene-responsive factor (ERF) gene, *OsERF3*, can serve as a central switch. This positively regulates the upregulation of genes such as *OsMPK3*, *OsMEK3*, *OsWRKY53*, and *OsWRKY70*, significantly improving rice’s resistance to the SSB [66]. *OsLRR-RLK1* may act upstream of the MPK signaling pathway, enhancing rice’s resistance to the SSB by positively regulating JA signaling and TrypPI activity [52]. In addition, in the rice variety, Minghui63, the upregulation of JA signaling genes, *OsLOXL-2*, *OsAOS3*, *OsJAZ1*, *OsJAZ7*, and *OsJAZ9*, contribute to the response of rice against the SSB [76].

### 3.2. JA Biosynthesis-Related Genes

According to reports, the JA signaling pathway plays a central role in defense responses across various plant species, including rice [17,120,121]. Studies have revealed that the oxylipins pathway gene, *OsHI-LOX*, provides a substrate for JA biosynthesis, positively enhancing rice’s resistance to the SSB [62]. The functional deficiency of *OsHPL3* in the lipid hydroperoxide lyase (HPL) gene is conducive to competition for more substrate—hydroperoxylinolenic acid (also required by HPL) by allene oxide synthase (AOS), thereby synthesizing more JA and improving the SSB resistance of rice [80]. After *Osr9-LOX1* in the 9-LOX pathway gene in rice was silenced, upon attack by the SSB, the content of linolenic acid (LeA) and JA increased significantly. At the same time, the content of the trypsin inhibitor also increased significantly, thereby inhibiting the growth of SSB larvae and greatly reducing damage to plants [77]. Research has found that overexpression of the allene oxide cyclase (*AOC*) gene and the OPDA (*cis*-12-oxo-phytodienoic acid) reductase (*OPR3*) gene in Zhonghua 11 enhances the resistance of rice plants to the SSB [78]. Furthermore, JA biosynthesis genes such as *OsLOX9*, *OsJAR1*, and *OsDAD1* were significantly induced under SSB infection [76]. Future investigations should further elucidate the role of these genes in mediating JA biosynthesis and conferring resistance to the SSB.

### 3.3. Plant Protease Inhibitor (PPI)

The plant protease inhibitor is one of the important defense substances of plants. It is generally a polypeptide or protein with a small molecular weight, which forms a complex with the protease in the insect’s digestive tract, blocking or weakening the hydrolysis of protease to protein in food. This makes insects anorexic or causes indigestion, resulting in death [101,122,123]. The plant protease inhibitor is generally induced through expression in plants. When some plants are attacked by insects, they respond by producing an oligosaccharide pheromone—protease inhibitor-inducing factor (PIIF) at the site of injury. This will induce the local production of plant protease inhibitor in the leaves and stimulate the production of signal substance—system peptide. Then, jasmonic acid will be generated through the action of a series of enzymes through the octadecanoic acid pathway to bind with the receptor, activating the plant protease inhibitor gene. After the gene, *mwti1b*, encoding winged bean trypsin protease inhibitor WTI-1B was introduced into rice plants for expression, the SSB larvae fed by transgenic rice plants showed significant growth delay, and its protein extract was proved to have an inhibitory effect on the intestinal protease of the SSB in vitro [106]. The expression of the maize protease inhibitor (*mpi*) gene in rice plants enhanced rice’s resistance to the SSB, which was specifically manifested in the significant weight reduction of the SSB larvae fed on *mpi* rice in a dose-dependent manner [108]. Next, researchers fused the maize protease inhibitor (MPI) (an inhibitor of insect serine proteinases) gene and the potato carboxypeptidase inhibitor (PCI) gene into an open reading frame and introduced it into rice plants. The described *mpi-pci* rice represents a more appropriate strategy for pest control than a strategy based on the use of a single PI gene by preventing the adaptive response of the SSB [101]. More importantly, after overexpression of the *AvrPiz-t* interacting protein 4 (*APIP4*) gene in rice, the accumulation of trypsin protease inhibitor increased, and the weight of the SSB fed with *APIP4* strains significantly decreased (compared with WT), while the performance of knockout strains (*apip4-5*) was the opposite [76]. In addition, the accumulation and activity level of trypsin plays a central role in all studies that have shown enhanced resistance of rice to the SSB [52,62,65,66,67,77,79,80].

### 3.4. Host-Induced Gene Silencing (HIGS)

HIGS is a strategy based on RNA interference (RNAi), involving the expression of appropriate RNAi constructs targeting insect genes in host plants, transferring double-stranded RNA (dsRNA), small interfering RNA (siRNA), or miRNA into insects during an interaction, and then silencing target genes and inhibiting or killing insects. Baum et al. [124] elucidated HIGS targeting pests for the first time in a landmark paper. They obtained transgenic maize expressing dsRNA designed for the ATPase A subunit of the western corn root worm (*Diabrotica virgifera*) for the first time, and found that the larvae of the western corn root worm died after feeding on the transgenic maize plant. The good insect resistance effect has demonstrated the potential of dsRNA transgenic plants in the field of crop pest control. Subsequent research has mostly adopted similar methods and attempted to develop RNAi crops targeting various insect species. Research has shown that overexpression of the SSB novel microRNA (miRNA) candidate gene, *csu-novol-miR15*, in rice (obtaining transgenic rice *csu-15*) will inhibit the growth of the SSB larvae feeding on the rice and delay the pupation stage [110]. An insect-specific and predicted target for Spook (*Spo*) and ecdysone receptor (*EcR*) in the insect ecdysone signaling network, miR-14, was found in the SSB and expressed in rice, resulting in transgenic rice exhibiting high resistance to the SSB [113]. SSB endogenous miRNA *csu-novol-miR260* negatively regulates ecdysteroid biosynthesis in the SSB by inhibiting the expression of *dib*. After being introduced into rice through amiRNA expression technology, rice exhibits high resistance to the striped stem borer [125]. In the same year, researchers also pointed out that the expression of miRNA *csu-novol-260* of the SSB in rice significantly inhibited the expression level of the *Csdib* gene of the SSB that feeds on it, showing significant resistance to the SSB and no cross-resistance to the resistant SSB for *Cry1C* rice [114]. Based on RNAi technology, the double-stranded RNA of the heat shock protein gene (*CssHsp*) of the SSB was transformed into rice. Bioassays showed that the transgenic lines (DS10, DS35, DS36) had a significant negative impact on the SSB population; after 8 days of feeding, the mortality rate of the three transgenic lines exceeded 60%. Through pupation, the mortality rate further increases to 90%, and very few SSBs survive until emergence [116]. SSB-resistant rice (*csu-53*) is produced by expressing SSB endogenous miRNA (*csu-novel-miR53*) based on RNAi technology. Feeding experiments have shown that *csu-53* rice inhibits the growth of SSB larvae delays pupation time and reduces the weight and emergence rate of SSB pupae [115]. The latest research has found that the double-stranded RNA gene expressing the fatty acyl CoA reductase (FAR) of SSB in rice shows a high level of resistance to the SSB, and the mortality rate of the SSB after feeding can reach 80% [117].

### 3.5. Bacillus thuringiensis (Bt) Gene

*B. thuringiensis* is a gram-positive soil bacillus that produces one or more insecticidal crystal proteins when forming spores. After being eaten by insects, insecticidal crystal proteins can bind to specific receptors on the intestinal epithelial cells of insects through the action of digestive enzymes in the intestine, ultimately leading to insect death. Since its discovery, *Bt* has been highly favored by plant protection workers due to its high insecticidal activity against target pests and its non-toxic properties against other non-target organisms such as natural enemies, mammals, and birds [126]. Since Belgian scientists first introduced the *Bt* gene into tobacco in 1987, research on *Bt* transgenic insect-resistant crops has developed rapidly. Li et al. [127] conducted field investigations and found that compared with the control variety, the occurrence of SSB larvae on Huahui No.1 (transgenic with *cry1Ab/1Ac* fusion gene) significantly decreased, with a decrease of 89.4% to 100%. Wang et al. [128] studied the *Bt* protein expression levels and insecticidal effects of three *Bt* rice varieties, Huahui No.1 (transgenic with *cry1Ab/1Ac* fusion gene), T1C-19 (transgenic with *cry1C* gene), and T2A-1 (transgenic with *cry2A* gene), at different stages through indoor bioassay. The research results showed that all three *Bt* rice varieties could produce a high mortality rate against the important rice pest, *C. suppressalis*, throughout the entire period. The fusion expression of *Cry1Ab* and *Vip3A* in rice shows high levels of resistance to the SSB and rice leaf folder [111]. Scientists hybridized four single *Bt* gene rice lines, *Cry1Ab*, *Cry1Ac*, *Cry2A*, and *Cry1C*, to obtain homozygous aggregated double *Bt* gene rice lines in the F3 generation. Laboratory bioassays showed that the double *Bt* gene line had higher activity against the SSB than its parents and exhibited superior resistance to the SSB in field evaluations [109]. Moreover, *Bt* rice can also serve as a dead-end trap crop for *C. suppressalis*, thereby protecting adjacent non-*Bt* rice plants from harm [112].

### 3.6. Other Ways

As early as 2001, researchers found that introducing the spider insecticidal gene, *SpI*, into rice enhanced its resistance to the SSB and rice leaf roller [107]. Li et al. [129] found that the promoter of the rice hydroperoxide lyase gene, *OsHPL2*, was specifically induced by the SSB, and then optimized to connect with *Cry1C* and transformed it into rice. Compared with the rice transformed with *Cry1C* alone, it showed a stronger level of resistance to the SSB. In addition, the deletion of the rice mitochondrial outer membrane protein 64 (*OM64*) gene also enhanced rice’s resistance to the SSB [61].

## 4. Conclusions and Prospects

In recent years, there is no doubt that the SSB has become a major threat to rice cultivation worldwide. The striped stem borer’s boring behavior and harmful characteristics of transformation, as well as its ability to attack various plants, make it a difficult-to-control pest. The induced defense response of rice is a complex process that is programmed to commence once the plant has identified the damage caused by the herbivorous insect. This process includes three parts: rice recognition of pest signals, early events of rice response to pest signals, and rice defense responses induced by early events. This article reviews the research progress in recent years on the early and mid-term events of rice response to SSB damage (such as MAPK cascades, JA, SA signals, etc.) and the production of downstream defense substances (such as TrypPIs, etc.). It can be seen that substantial progress has been made in this field of research. In addition, strategies for the heterologous expression of genes from other species in rice to combat the SSB were reviewed, especially the HIGS strategy, which has currently attracted much research attention. However, compared with the research on the three major diseases of rice and pests such as the brown planthopper, there are still many urgent problems and directions for further efforts in the research into rice’s resistance to the SSB.

Firstly, there is insufficient research on the recognition of SSB hazard signals in rice plants, including plant pattern recognition receptors (PRRs) and herbivore-associated molecular patterns (HAMPs). Currently, only the leucine-rich repeat receptor-like kinase 1 (LRR-RLK1) has been shown to play an important role in rice’s recognition of the SSB [52]. In terms of the interaction between rice and the BPH, research has revealed that three G-type lectin receptor kinases (LecRKs) can enhance rice’s ability to resist the BPH [15]. The C-terminal region of mucin secreted by the BPH can induce a defense response of rice to the BPH [130]. The BPH’s salivary protein, BISP, can be directly recognized by BPH14, triggering effector-triggered immunity in rice [12]. Therefore, future research should focus on the identification of new HAMPs and rice plant receptors. In addition, further exploration of the interaction patterns between HAMPs and receptors in rice–insect interactions and their identification can be considered, such as plant immunity triggered by pattern recognition receptors (PRR-triggered immunity, PTI) and effector-triggered immunity (ETI).

Although substantial progress has been made in the study of early events of plants’ response to pest signals in recent years, there is still little research on early events of rice’s response to chewing mouthpart insects (such as the SSB). From the current progress in research, it can be seen that the early events of plants’ response to pest signals exhibit a certain broad spectrum; that is, different pests induce the same early events after damaging plants. By utilizing this phenomenon, we can refer to the early signaling events of rice’s response to signals from other pests such as brown planthoppers to study the early signaling events of rice’s response to the SSB. Research has found that silencing *OsERF3* [66] and *OsWRKY70* [131], or overexpressing *OsWRKY53* [132] in rice can significantly increase the accumulation of H_2_O_2_ induced by the BPH, thereby enhancing the resistance of transgenic plants to the BPH. *OsMKK4* can activate *OsMPK3/OsMPK6*, thereby positively regulating the JA signaling pathway and enhancing resistance to pests [131]. At the same time, there are complex interactions between early signaling events, such as MAPK cascade signaling and Ca^2+^ signaling pathways, which rely on plant pattern recognition receptors [133]. The Ca^2+^ signaling pathway is widely involved in other early signal events, including the Vm depolarization process and the regulation of NADPH oxidase activity, thereby regulating the burst of reactive oxygen species [134,135,136]. Therefore, there is still a long way to go to comprehensively and fully understand the entire regulatory network of early signaling events in rice’s defense response to the SSB.

In the study of defense compounds against the SSB, only plant protease inhibitors (which reduce the insect’s ability to digest and absorb nutrients) and green leaf volatiles (which attract natural enemies to attack target pests) have been extensively studied, and therefore, more research is needed. Research has shown that tannins, as a toxic substance, can bind to various enzyme proteins in insects to inhibit enzyme activity, ultimately reducing insect survival ability [137]. Plant ecdysteroid (a steroidal compound) has a similar function to ecdysteroid in insects. Low doses can promote insect ecdysis, affecting their normal growth and development [97]. After entering the midgut of insects, plant lectin combines with glycoconjugates, making it difficult for pests to digest and absorb food, thus inhibiting their growth and development [138]. The overexpression of polyphenol oxidase synthesis genes in tomatoes can significantly enhance their resistance to *Spodoptera litura* [139]. In addition, protease inhibitors can appear as auxiliary protective proteins in plant defense reactions against insects, protecting defense proteins that are easily hydrolyzed [140]. More importantly, terpenoids, flavonoids, alkaloids, peroxidase, chitinase, and leucine aminopeptidase are all defense substances of plants against insect attacks. Therefore, in the future, more defense compounds (rice against the SSB) need to be excavated and identified.

Rice borer resistance is a quantitative trait that, like most important agronomic traits such as yield, quality, maturity, and drought resistance, is controlled by micro polygenes, namely, quantitative trait genes or quantitative trait loci (QTLs), manifested as quantitative inheritance. At present, the cloning and functional verification of key regions of SSB-resistant QTLs in rice have not been reported. Fortunately, with the progress made in sequencing technology, it has become easier to obtain huge genome resources. Through the study of genome-wide association, a large number of genes significantly related to insect resistance phenotypes can be mined. Based on this, our research team has sequenced the whole genome of 258 rice varieties, established models, and conducted association analysis with insect resistance traits. As a result, we have obtained multiple associated genes and transformed rice for functional verification. In addition, small RNAs (including miRNAs, piRNAs, tsRNAs, snRNAs, and snoRNAs) are a class of RNA molecules that do not have protein-coding capabilities and can regulate gene expression. They play an important role in basic biological processes such as cell growth, development, and metabolism [141]. Epigenetics mainly includes DNA modification, histone modification, and post-translational modifications (PTMs), etc. These modifications cause changes in the spatial structure or chromatin structure of DNA, leading to heritable changes in gene function [141]. Therefore, small RNA sequencing and the exploration of epigenetic mechanisms will be important directions for future research. In summary, we hope that, with the discovery and utilization of new resistance sources, the gradual deepening of resistance genetic research, the wide application of biotechnology and molecular genetics in agricultural production, and the close cooperation and collaboration of researchers, the breeding of insect-resistant varieties will reach established goals. This will further improve the quality and yield of rice, which will, in turn, improve people’s living standards.

## Figures and Tables

**Figure 1 ijms-24-14361-f001:**
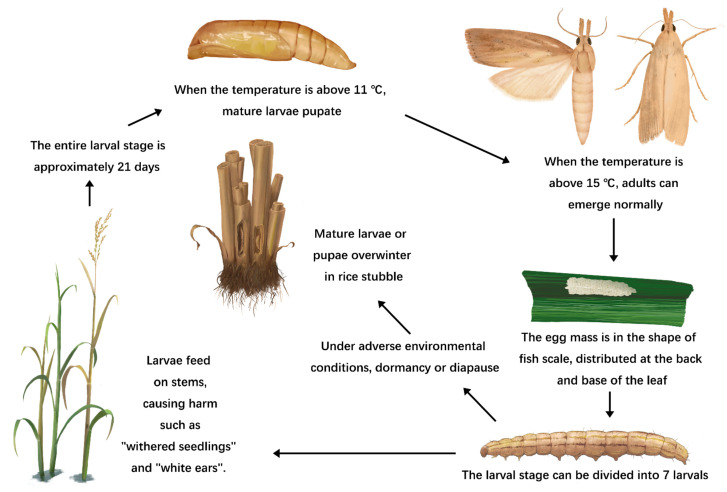
Generation cycle of *C. suppressalis* damage on rice.

**Figure 2 ijms-24-14361-f002:**
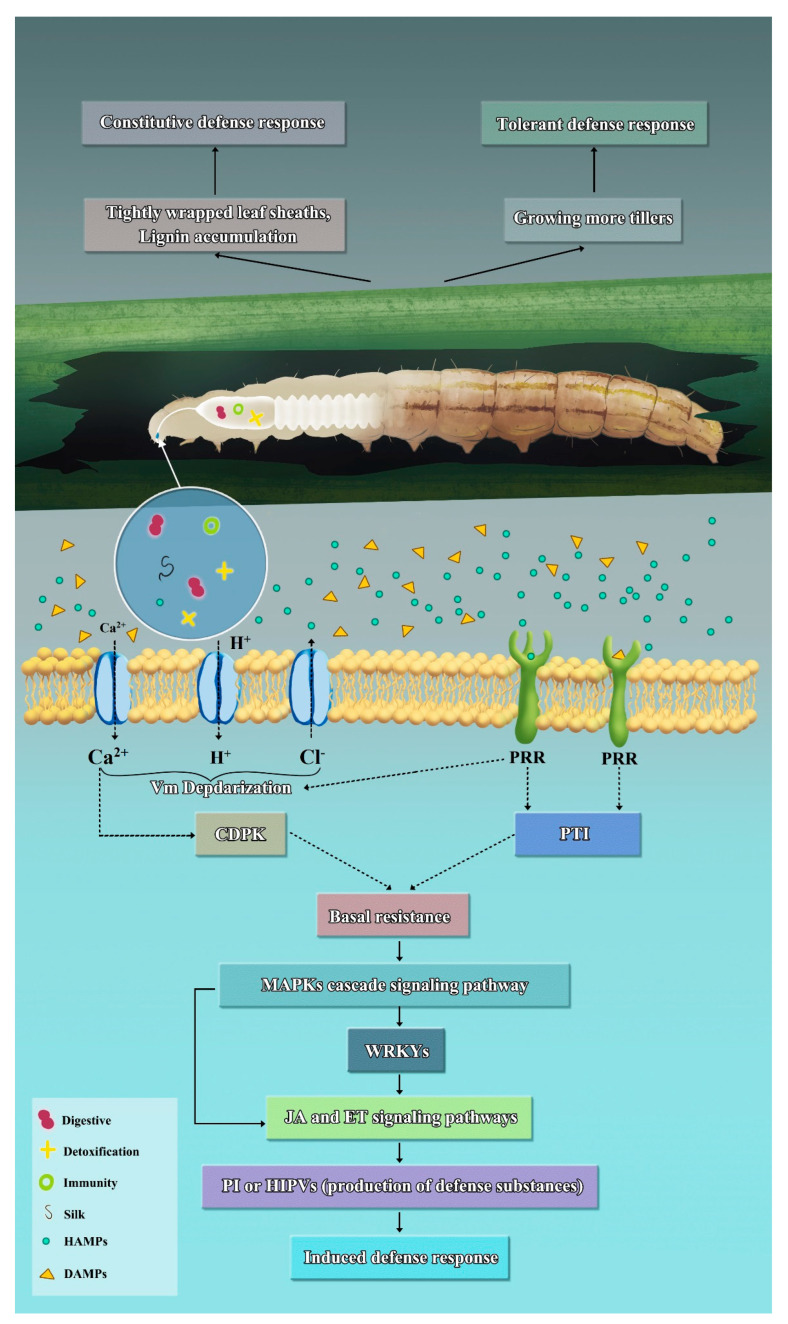
Preliminary model of rice defense strategy against the SSB.

**Table 1 ijms-24-14361-t001:** Transgenic rice against *C. suppressalis*: a historical overview.

Rice Cultivar	Transformation Method	Type	Gene	Origin	Comments	References
Nipponbare	*Agrobacterium* mediated	Overexpression	*mwti1b*	Winged bean	Trypsin inhibitor	Mochizuki et al. (1999) [106]
Xiushui 11, Chunjiang 11	*Agrobacterium* mediated	Overexpression	*SpI*	Spider	Spider insect toxin	Huang et al. (2001) [107]
Senia, Ariete	particle bombardment and *Agrobacterium* mediated	Overexpression	*mpi*	Maize	Proteinase inhibitor	Vila et al. (2005) [108]
Xiushui 11	*Agrobacterium* mediated	Antisense expression	*OsHI-LOX*	Rice	Type 2 13-lipoxygenase	Zhou et al. (2009) [62]
Xiushui 110	*Agrobacterium* mediated	Overexpression	*OsERF3*	Rice	Ethylene-responsive factors	Lu et al. (2011) [66]
Minghui 63	*Agrobacterium* mediated	Overexpression	*Cry1Ab, Cry1AC, Cry1C, Cry2A*	Bt strains	Insecticidal crystal protein	Yang et al. (2011) [109]
Zhonghua 11	γ-rays	Mutation	*OsHPL3*	Rice	A hydroperoxide lyase	Tong et al. (2012) [80]
Xiushui 11	*Agrobacterium* mediated	RNAi	*OsMPK3*	Rice	Mitogen-activated protein kinases	Wang et al. (2013) [65]
Zhonghua 11	*Agrobacterium* mediated	Overexpression	*OsAOC*	Rice	Aladiene oxide cyclase	Guo et al. (2014) [78]
Zhonghua 11	*Agrobacterium* mediated	Overexpression	*OsOPR3*	Rice	*Cis*-12-oxo-phytodienoic acid reductase 3	Guo et al. (2014) [78]
Rice cultivar	Transformation method	Type	Gene	Origin	Comments	References
Xiushui 11	*Agrobacterium* mediated	Antisense expression	*Osr9-LOX1*	Rice	9-lipoxygenase	Zhou et al. (2014) [77]
Ariete	*Agrobacterium* mediated	Overexpression	*mpi, pci*	Maize, Potato	Maize proteinase inhibitor, potato carboxypeptidase inhibitor	Quilis et al. (2014) [101]
Xiushui 11	*Agrobacterium* mediated	RNAi	*OsWRKY53*	Rice	Transcription factor	Hu et al. (2015) [67]
Zhonghua 11	*Agrobacterium* mediated	HIGS	*csu-novel-miR15*	SSB	Insect endogenous small RNAs	Jiang et al. (2016) [110]
Xiushui 134	*Agrobacterium* mediated	Overexpression	*Cry1Ab, Vip3A*	Bt strains	Insecticidal crystal protein	Xu et al. (2018) [111]
Minghui 63	*Agrobacterium* mediated	Overexpression	*Cry1C*	Bt strains	Insecticidal crystal protein	Jiao et al. (2018) [112]
Xidao NO1	Lipofectin transfection	knocking out	*CYP71A1*	Rice	Tryptamine 5-hydroxylase	Lu et al. (2018) [16]
Zhonghua 11	*Agrobacterium* mediated	HIGS	*miR-14*	SSB	Insect endogenous small RNAs	He et al. (2019) [113]
Zhonghua 11	*Agrobacterium* mediated	Overexpression	*OsHPL2 promoter and Cry1C*	Rice and Bt strains	SSB-inducible promoter of rice gene, *OsHPL2*	Li et al. (2020) [7]
Zhonghua 11	*Agrobacterium* mediated	T-DNA insertion	*om64*	Rice	rice mitochondrial outer membrane protein 64	Guo et al. (2020) [61]
Rice cultivar	Transformation method	Type	Gene	Origin	Comments	References
Nipponbare	*Agrobacterium* mediated	Overexpression	*APIP4-OX-16-2*	Rice	the Bowman–Birk inhibitor *AvrPiz-t* interacting protein 4	Liu et al. (2021) [76]
Zhonghua 11	*Agrobacterium* mediated	HIGS	*csu-novel-260*	SSB	Insect endogenous small RNAs	Zheng et al. (2021) [27] Wen et al. (2021) [114]
Zhonghua 11	*Agrobacterium* mediated	HIGS	*csu-novel-miR53*	SSB	Insect endogenous small RNAs	Liu et al. (2022) [115]
Zhonghua 11	*Agrobacterium* mediated	HIGS	*CssHsp*	SSB	Small heat shock protein	Mao et al. (2022) [116]
Zhonghua 11	*Agrobacterium* mediated	HIGS	*CsFAR*	SSB	Fatty acyl-CoA reductase	Sun et al. (2022) [117]

## Data Availability

Not applicable.

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
