# Peer review of "Defense Strategies of Rice in Response to the Attack of the Herbivorous Insect, Chilo suppressalis"

_ijms, 2023, doi:10.3390/ijms241814361_

Round 1

Reviewer 1 Report

The review comprehensively describes all the information known to date on the molecular genetic response to SSB contamination.  The review is well structured, written in clear language.

Disadvantages:

1. The word "end" should be deleted from the list of authors

2. Speaking about non-native rice genes and strategies to counteract SBB (pages 13-14), the authors, unfortunately, do not talk about the possible impact of products of these genes on the human body and their safety.

Reviewer 2 Report

Comments, 

1)    What do you mean by boring characteristics here? Can you use such phrases in scientific literature. 

2)    Line 16. Why J is capital here?

3)    How about crispr cas9 related studies? Include them.

4)    How about other omics studies? small RNA sequencing or metabolomics related studies. Include them. 

5)    How about epigenetic mechanisms? Include such studies.

6)    Enhancing comprehension and integrating more detailed information into the discussion are necessary.

7)    Major editing of English language is needed. 

  Major editing of English language is needed. 
